# Risk assessment and antibiotic prescribing decisions in children presenting to UK primary care with cough: a vignette study

Martine Nurek ![ORCID] , Brendan C Delaney, Olga Kostopoulou

Surgery and Cancer, Imperial College London, London, UK

**Correspondence to**
Dr Martine Nurek;
m.nurek@imperial.ac.uk

## ABSTRACT

**Objectives** The validated 'STARWAVe' (Short illness duration, Temperature, Age, Recession, Wheeze, Asthma, Vomiting) clinical prediction rule (CPR) uses seven variables to guide risk assessment and antimicrobial stewardship in children presenting with cough. We aimed to compare general practitioners' (GPs) risk assessments and prescribing decisions to those of STARWAVe and assess the influence of the CPR's clinical variables.

**Setting** Primary care.

**Participants** 252 GPs, currently practising in the UK.

**Design** GPs were randomly assigned to view four (of a possible eight) clinical vignettes online. Each vignette depicted a child presenting with cough, who was described in terms of the seven STARWAVe variables. Systematically, we manipulated patient age (20 months vs 5 years), illness duration (3 vs 6 days), vomiting (present vs absent) and wheeze (present vs absent), holding the remaining STARWAVe variables constant.

**Outcome measures** Per vignette, GPs assessed risk of hospitalisation and indicated whether they would prescribe antibiotics or not.

**Results** GPs overestimated risk of hospitalisation in 9% of vignette presentations (88/1008) and underestimated it in 46% (459/1008). Despite underestimating risk, they overprescribed: 78% of prescriptions were unnecessary relative to GPs' own risk assessments (121/156), while 83% were unnecessary relative to STARWAVe risk assessments (130/156). All four of the manipulated variables influenced risk assessments, but only three influenced prescribing decisions: a shorter illness duration reduced prescribing odds (OR 0.14, 95% CI 0.08 to 0.27, p<0.001), while vomiting and wheeze increased them (OR$_{vomit}$ 2.17, 95% CI 1.32 to 3.57, p=0.002; OR$_{wheeze}$ 8.98, 95% CI 4.99 to 16.15, p<0.001).

**Conclusions** Relative to STARWAVe, GPs underestimated risk of hospitalisation, overprescribed and appeared to misinterpret illness duration (prescribing for longer rather than shorter illnesses). It is important to ascertain discrepancies between CPRs and current clinical practice. This has implications for the integration of CPRs into the electronic health record and the provision of intelligible explanations to decision-makers.

---

### Strengths and limitations of this study

► This is the first study to suggest discrepancies between the STARWAVe (Short illness duration, Temperature, Age, Recession, Wheeze, Asthma, Vomiting) clinical prediction rule and current clinical practice.

► Use of clinical vignettes allowed us to manipulate some variables while holding others constant; thus we could identify *causal* relationships between specific clinical variables and antibiotic prescribing decisions.

► In so doing, we bring much-needed experimental evidence to the literature, which is currently dominated by interview and observational studies.

► The disadvantage of using clinical vignettes is that our results are based on hypothetical clinical scenarios, which contained limited information.

► Moreover, we manipulated only a subset of the STARWAVe variables; future work could increase the number of clinical variables manipulated and explore non-clinical factors too.

---

## INTRODUCTION

Combatting antimicrobial resistance is high on policy agendas internationally.[1–3] One of the key means advocated is judicious antibiotic prescribing.[1] Over 80% of all National Health Service antibiotic prescriptions are issued in primary care,[4] where despite numerous campaigns, mandates and financial incentives, rates remain unacceptably high.[5] Despite strong evidence of only modest symptomatic benefits for acute respiratory tract infections (RTIs),[6–8] and even smaller effects against complications,[9 10] RTIs are the most common justification for primary care antibiotic use[11] and a leading cause of overuse.[12] This is exacerbated in children, where perceived vulnerability and prognostic uncertainty (ie, perceived risk of deterioration) can lead to defensive prescribing ('treat, just in case').[12–15]

To improve risk assessment and antimicrobial prescribing in children with RTIs, a clinical prediction rule (CPR) called 'STARWAVe' (Short illness duration, Temperature, Age, Recession, Wheeze, Asthma, Vomiting) was recently developed and validated.[12] It was based on a large prognostic cohort study, which included 8394 children presenting to 247 general practices in England with acute cough and RTI symptoms.[12] Numerous characteristics were recorded at presentation, including demographic variables, parent-reported symptoms and physical examination signs. In a regression analysis, seven of these characteristics were found to predict hospital admission (for RTI) in the month following presentation: Short illness duration (≤3 days), Temperature (≥37.8°C), Age (<2 years), Recession, Wheeze, Asthma and Vomiting.[12] This analysis gave rise to the 'STARWAVe' clinical prediction rule: a seven-item, point-of-care checklist that can distinguish children at 'very low' (0.3%, with ≤1 characteristic), 'normal' (1.5%, with 2 to 3 characteristics) and 'high' (11.8%, with ≥4 characteristics) risk of hospitalisation, with good accuracy (area under the receiver operating characteristic curve 0.81, 95% CI 0.76 to 0.85).[12] Using STARWAVe, clinicians can quickly and reliably identify the 'high risk' cases that might warrant antimicrobial treatment. More importantly, they can identify the 'very low risk' and 'normal risk' cases that will likely resolve on their own, and spare them unnecessary treatment.[12]

STARWAVe is thus a prognostic (not a diagnostic) tool. It cannot tell clinicians whether an infection is bacterial or viral. This does not however invalidate it as an antimicrobial prescribing aid because overprescribing is so often driven by prognostic concerns.[12–15] STARWAVe recognises this and addresses it, by providing evidence-based reassurance (to clinicians and perhaps even parents) that specific children are *not* at significant risk. In so doing, it can assuage the fears and anxieties that are known to trigger unnecessary prescriptions.

Like other CPRs and clinical risk scores (eg, QCancer), STARWAVe could be integrated into the electronic health record to guide clinicians' risk assessments and prescribing decisions. In fact, one research group has incorporated web-based STARWAVe decision support into a multifaceted intervention that aims to improve the management of children presenting with cough in primary care (the intervention is currently undergoing clinical trial).[16] As a rule, decision support should be transparent and intelligible to the decision-maker;[17] a risk score is merely a probability and could be ignored, especially if it contradicts the decision-maker's intuitive assessment of risk.[18] Thus, it is important to understand whether and how general practitioners' (GPs) intuitive risk assessments and prescribing decisions differ from those of STARWAVe and how GPs interpret the CPR's clinical variables.

To explore this, we presented GPs with clinical vignettes describing children presenting with cough. The vignettes included all seven STARWAVe variables; however, only four were manipulated (ie, varied systematically across the vignettes). This was due to logistical constraints: these data were collected in conjunction with another study,[19] which limited the number of vignettes that we could present and thus the number of variables that we could manipulate. We chose to manipulate patient age (20 months vs 5 years), illness duration (3 days vs 6 days), vomiting (present vs absent) and wheeze (present vs absent), holding the remaining STARWAVe variables constant (temperature, asthma and recession). Fever was present in all of the vignettes, as it is a common presenting feature of childhood RTIs.[12] Asthma and recession are both associated with airflow obstruction, but wheeze (another symptom of airflow obstruction) was more common in the STARWAVe cohort;[12] therefore we chose to manipulate wheeze and kept asthma and recession constant across vignettes (always absent). Per vignette, GPs assessed risk of hospitalisation (very low, normal or high) and indicated whether they would prescribe antibiotics or not. We compared GPs' intuitive risk assessments and prescribing decisions to STARWAVe guidelines and assessed the influence of the manipulated STARWAVe variables.

## METHOD
### Participants
#### Sample size
In the STARWAVe elicitation and validation study, a young age (<2 years), a short illness duration (≤3 days), vomiting (present vs absent) and wheeze (present vs absent) were found to increase the odds of hospitalisation twofold to threefold (OR range 2.16 to 3.42; all p values ≤0.004).[12] We powered the present study to detect effects of the same size on the decision to prescribe antibiotics. Specifically, using G*Power 3.1, we estimated that in order to detect the smallest effect (OR 2.16) in a two-tailed logistic regression of prescribing (yes vs no) on the four manipulated factors (with power=80% and α=0.05), 226 responses would be required.

#### Recruitment
By email, we invited certified and practising UK GPs that had participated in previous studies by our research group. In addition, the NIHR-CRN (National Institute for Health Research Clinical Research Network) circulated our invitation email to general practices across England.

#### Design and materials
Study materials were eight clinical vignettes that depicted children presenting to the GP with cough. Each child was described in terms of the seven STARWAVe variables. In a $2^{4-1}$ fractional factorial design, we manipulated patient age (20 months vs 5 years), illness duration (3 days vs 6 days), vomiting (present vs absent) and wheeze (present vs absent), holding the remaining variables constant (presence of fever, absence of asthma and recession). We chose to use a fractional factorial design (rather than a full factorial design) because it delivers clear estimates

of main effects using half the number of vignettes (ie, 8 rather than 16).[20]

Risk of hospitalisation ranged from 'very low' (vignette 1 in online supplementary appendix 1) to 'high' (vignette 8 in online supplementary appendix 1), but in most cases it was 'normal' (vignettes 2 to 7 in online supplementary appendix 1). Thus, only one vignette warranted a prescription according to STARWAVe (vignette 8). Each participant was randomly assigned to view four of the eight vignettes.

## Procedure

Interested participants were emailed a link to the study website where they read an information sheet and provided informed consent. Thereafter, they saw 26 clinical vignettes: 2 pertained to this study and 24 pertained to an unrelated study conducted by our research group, concerning referral for suspected cancer.[19] The two antibiotics vignettes were presented after 33% and 66% of the cancer vignettes, respectively, and were introduced as follows: 'We understand that this is somewhat monotonous, so here is something quite different to help you re-engage attention'. The antibiotics and cancer vignettes were comparable in length and difficulty.

Twenty-four hours after completing this questionnaire, participants were emailed a link to a second questionnaire, which was structured in the same way; that is, two antibiotics vignettes were evenly dispersed among 24 cancer vignettes. Importantly, the four antibiotics vignettes seen by a given participant were selected at random and presented in a random order.

Following each antibiotics vignette, GPs were asked two questions:

▶ *In your opinion, what is the risk that this child would deteriorate, requiring hospital admission?*
  – *very low risk, for example, 1 in 300*
  – *medium risk, for example, 1 in 70 (in STARWAVe, this level of risk is labelled 'normal')*
  – *high risk, for example, 1 in 8*
▶ *In your clinical judgement, what would be the best course of action?*
  – *no antibiotics prescription*
  – *antibiotics prescription*
  – *delayed antibiotics prescription*

A delayed antibiotics prescription is a forward-dated prescription, intended for use by the patient if symptoms do not improve by the specified date. Delayed prescriptions form part of the national strategy to reduce immediate prescribing.[21] They were not the focus of the present study, but were included to ensure that the options available were representative of daily practice, and that our measure of immediate prescribing was precise (ie, not skewed by the absence of an option that is typically present).

Twenty-four hours later, participants were emailed a link to a third questionnaire; specifically, Gerrity *et al*'s Stress from Uncertainty scale, which is one of the Physicians' Reactions to Uncertainty scales.[22] The Stress from Uncertainty scale is a self-report measure of the extent to which physicians experience anxiety due to clinical uncertainty and concern about bad outcomes.[22] We expected that GPs who experience greater Stress from Uncertainty (SfU) would also experience greater prognostic uncertainty when assessing children with RTIs, and thus be more inclined to prescribe. GPs were asked to indicate their agreement with each of the scale's eight items (presented in a random order) on a 6-point Likert scale anchored at 1='strongly disagree' and 6='strongly agree' (online supplementary appendix 2).

## Analyses

To investigate the effect of the manipulated factors on risk assessments and prescribing decisions, two logistic regression models were built. The first was an ordinal logistic regression model, where patient age (0=5 years, 1=20 months), illness duration (0=6 days, 1=3 days), vomiting (0=absent, 1=present) and wheeze (0=absent, 1=present) were used to predict perceived risk of hospitalisation (0=very low, 1=medium, 2=high). The second was a binary logistic regression model, where the same independent variables were used to predict prescribing decisions (0=no prescription, 1=prescription), which we dichotomised by merging 'no prescription' and 'delayed prescription' into a single category (national guidelines for antimicrobial prescribing treat them interchangeably[21]). For the interested reader, results pertaining to delayed prescriptions are presented in online supplementary appendix 3.

In two further logistic regression models (one ordinal and one binary), we investigated whether SfU scores (summed across items per GP) might relate to risk assessments (0=very low, 1=medium, 2=high) and prescribing decisions (0=no prescription, 1=prescription).

Statistical analysis was performed using Stata/MP 13.1. Specifically, the ordinal analyses were conducted using the Stata user-written programme 'gologit2',[23 24] where we computed cluster-robust standard errors to account for repeated measures (multiple responses per GP). The binary analyses were conducted using Stata's 'melogit' command,[25] where we included a random intercept for GPs.

## Patient and public involvement

Patients and members of the public were not involved in the design, execution, reporting or dissemination of this research.

## RESULTS
### Descriptive statistics

We collected data from 254 GPs. Of these, two gave only partial data and thus were excluded from the analyses. The final sample comprised 252 GPs, with an average of 15 years' experience in general practice post-qualification (*SD* 9.8). Half of the sample was female (52%, 131/252). Eighty-six per cent were recruited via direct email from

**Table 1** Association between risk as classified by GPs and as classified by STARWAVe

| STARWAVe risk | Risk as classified by GPs | | | Total |
|---|---|---|---|---|
| | Very low | Medium | High | |
| Very low | 81 | 44 | 1 | 126 |
| Medium ('normal') | 345 | 368 | 43 | 756 |
| High | 33 | 81 | 12 | 126 |
| **Total** | 459 | 493 | 56 | 1008 |

GPs, general practitioners.

the research team (217/252) and 14% via the NIHR-CRN (35/252).

Each GP saw four vignettes, yielding 1008 case presentations. GPs correctly classified risk of hospitalisation in 46% of these (461/1008; table 1). Risk was rarely overestimated (9% of responses, 88/1008; blue cells) but frequently underestimated (46% of responses, 459/1008; green cells). Specifically, medium risk patients were classified as very low risk 46% of the time (345/756), while high risk patients were classified as very low or medium risk 90% of the time (114/126).

GPs classified risk as high only 6% of the time (56/1008) but prescribed immediately 15% of the time (156/1008), suggesting a dissociation between risk assessments and prescribing decisions. Indeed, 78% of prescriptions were not consistent with GPs' own risk assessments (121/156; table 2, blue cells) and 83% were not consistent with STARWAVe risk assessments (130/156; table 2, green cells).

Online supplementary appendix 4 presents the number and proportion of prescriptions per vignette. The case with the highest rate of prescription was not the high risk case, which received a prescription only 21% of the time (26/126; vignette 8). Rather, it was a medium risk case, describing a 5-year-old child with a 6-day illness duration who had both vomiting and wheeze (33%, 42/126; vignette 7).

### Results of planned analyses

Younger patient age (20 months vs 5 years) increased perceived risk of hospitalisation (OR 1.49, 95% CI 1.14 to 1.95, p=0.003), while a short illness duration decreased it (OR 0.54, 95% CI 0.42 to 0.69, p<0.001). Presence of vomiting and presence of wheeze were both associated with higher risk estimates ($OR_{vomit}$ 1.92, 95% CI 1.57 to

2.36, p<0.001; $OR_{wheeze}$ 3.33, 95% CI 2.66 to 4.16, p<0.001). Statistical tests of the proportional odds assumption revealed that all four variables met it; that is, the effect of each independent variable was consistent for successive levels of the ordinal dependent variable (all p values ≥0.099). A global Wald test confirmed that the proportional odds assumption was not violated in this model ($\chi^2$ (4) 4.70, p=0.320).

Patient age did not influence the odds of a prescription (OR 1.42, 95% CI 0.83 to 2.42, p=0.201), but a short illness duration decreased them (OR 0.14, 95% CI 0.08 to 0.27, p<0.001). Presence of vomiting and presence of wheeze both increased prescribing odds ($OR_{vomit}$ 2.17, 95% CI 1.32 to 3.57, p=0.002; $OR_{wheeze}$ 8.98, 95% CI 4.99 to 16.15, p<0.001). When prescribing was treated as a 3-category ordinal variable (0=no prescription, 1=delayed prescription, 2=immediate prescription), these findings did not change (online supplementary appendix 3).

SfU scores were unrelated to risk assessments (OR 1.00, 95% CI 0.98 to 1.02, p=0.935; proportional odds assumption met with $p_{SfU}$=0.406) and prescribing decisions (OR 1.00, 95% CI 0.96 to 1.03, p=0.875).

### DISCUSSION

We compared GPs' risk assessments and antimicrobial prescribing decisions to a normative model (the STARWAVe CPR), in the context of clinical vignettes that varied the features (age, illness duration, vomiting and wheeze) of children presenting with cough. Relative to STARWAVe, GPs frequently underestimated the patient's risk of deterioration, but nonetheless overprescribed: the vast majority of their prescriptions were unnecessary relative to their own risk assessments (78%) and STARWAVe risk assessments (83%).

This is not the first study to identify a dissociation between risk assessments and antimicrobial prescribing decisions. In one study, for example, an educational intervention was successful in reducing physicians' overestimations of the likelihood of a bacterial infection, but unsuccessful in reducing antibiotic prescribing.[26] In another, patient expectations for antibiotics increased physicians' rates of antibiotic prescribing, but did not influence their probability estimates of a bacterial infection.[27] Presently, a dissociation between risk assessments and antibiotic prescribing decisions suggests that the former may not be the sole determinant of the latter. It

**Table 2** Association between risk (as classified by GPs and by STARWAVe) and prescribing decisions

| Prescriptions | Risk as classified by GPs | | | STARWAVe risk | | | Total |
|---|---|---|---|---|---|---|---|
| | Very low | Medium | High | Very low | Medium ('normal') | High | |
| None/delayed | 445 | 386 | 21 | 112 | 640 | 100 | 852 |
| Immediate | 14 | 107 | 35 | 14 | 116 | 26 | 156 |
| **Total** | 459 | 493 | 56 | 126 | 756 | 126 | 1008 |

GPs, general practitioners.

**Table 3** The effect of patient age, illness duration, vomiting and wheeze on risk of hospitalisation, according to present participants ($OR_{GPs}$) and STARWAVe ($OR_{STARWAVe}$)

| Predictor | $OR_{GPs}$ | $OR_{STARWAVe}$ |
|---|---|---|
| Age (<2 years) | 1.49 [1.14 to 1.95] | 3.42 [2.12 to 5.58]* |
| Duration (≤3 days) | 0.54 [0.42 to 0.69]* | 2.77 [1.77 to 4.35]* |
| Vomiting | 1.92 [1.57 to 2.36]* | 2.56 [1.54 to 4.31]* |
| Wheeze | 3.33 [2.66 to 4.16]* | 2.16 [1.28 to 3.60]* |
| Temperature | | 1.99 [1.22 to 3.25]* |
| Asthma | | 3.93 [2.20 to 7.03]* |
| Recession | | 3.82 [2.23 to 6.62]* |

*p≤0.006. Square brackets contain 95% CIs.
GPs, general practitioners.

is also possible that explicit risk ratings (as elicited in this study) do not reflect physicians' intuitive assessments of risk.

All four of the manipulated variables influenced physicians' (explicit) risk assessments, which increased when the child was younger (20 months vs 5 years), when illness duration was longer (6 vs 3 days) and when vomiting and/or wheeze were present (vs absent). Comparing the ORs for these relationships to the STARWAVe model (table 3), we note both similarities and discrepancies. Specifically, GPs' interpretations of patient age, vomiting and wheeze were consistent with the STARWAVe model, but their interpretation of illness duration was not: a shorter illness duration reduced—rather than increased—GP estimates of risk.

Like risk assessments, prescribing increased when illness duration was long (inverted OR 7.14) and when vomiting and/or wheeze were present ($OR_{vomit}$ 2.17; $OR_{wheeze}$ 8.98). Patient age had no reliable effect on prescribing (OR 1.42). Again, these findings are not entirely consistent with the STARWAVe model, but they are consistent with previous, non-experimental research. In one interview study, for example, GPs reported that they were more likely to prescribe antibiotics to children with RTIs given prolonged duration of symptoms, abnormal chest signs and (less frequently) vomiting.[13] Various observational studies have likewise identified chest abnormalities[28–32] and vomiting[31] as clinical characteristics that prompt prescribing. In contrast, previous literature concerning the effect of age on prescribing is mixed: two studies found that older (vs younger) patients were more likely to receive a prescription,[30 33] but three identified no association between age and prescribing.[29 31 32]

Interestingly, the one patient that may have warranted a prescription received one only 21% of the time. This appears low, but in fact only 27% of hospitalised children in the STARWAVe cohort had a discharge diagnosis suggestive of a bacterial infection.[12] Consequently, STARWAVe does not argue (or prove) that all high risk children require immediate antimicrobial treatment; rather, it recommends close monitoring and urgent follow-up with a view to prescribe if needed.[12] Viewed thus, the rate of prescription that we observed in high risk cases (21%) seems not low, but well-calibrated to the epidemiological landscape (27%).

Risk assessments and prescribing tendencies bore no association to GPs' self-reported levels of 'Stress from Uncertainty'. However, Grol and colleagues found that greater willingness to take risks (as measured on their Attitudes to Risk Taking scale) was associated with significantly fewer antibiotics prescriptions for respiratory problems and upper respiratory tract infection/common cold.[34] Attitudes toward risk—rather than attitudes toward uncertainty—may thus prove a fruitful avenue for future research.

### Limitations and future work

This is the first study to identify discrepancies between the STARWAVe clinical prediction rule and current clinical practice. There are several possible reasons for these discrepancies. First, GPs may be unaware of the STARWAVe rule, which was published only 4 years ago; if so, then dissemination and training may be needed. Alternatively, GPs may be aware of the rule but fail to deploy it at the point of care; in this case, automated STARWAVe support (eg, incorporation of STARWAVe metrics into the electronic health record) could increase uptake. Even so, the rule is intended to "…supplement, not supplant, clinical judgment" (p. 908)[12] and thus—thirdly—GPs may choose to override it for sound clinical reasons. Consider, for example, that a long illness duration (6 vs 3 days) triggered prescriptions in the present study; this is inconsistent with STARWAVe, but could form part of GPs' strategy to reduce prescriptions, if the alternative is to prescribe early in the illness (ie, a 'wait-and-see' approach). Nonetheless, a more evidence-based strategy is not to prescribe at all in simple RTI, which is likely to last longer than 6 days in any case.[21 35] Finally, it is also possible that methodological aspects of the present study contributed to the discrepancies observed. For example, the distribution of risk in our vignettes (13% very low, 75% medium and 13% high) was not representative of the patient population (67% very low, 30% medium and 3% high[12])—an unavoidable consequence of our fractional factorial design. In the 'real world', GPs see many more very low risk cases (67% rather than 13%) and fewer medium and high risk cases (30% rather than 75% medium; 3% rather than 13% high). This may have hurt GPs' performance by being ecologically invalid (ie, mismatched to true base rates) and could explain their tendency to underestimate risk in the present study.

A more representative set of vignettes would enhance not only the external validity of the study but also the clinical significance of the findings. Our findings speak mostly to the medium risk group (because we employed mostly medium risk cases) but very low risk cases are twice as common in clinical practice and indeed account for two-thirds of child RTI presentations in primary care.[12] They are also the focal point of the STARWAVe rule, which aims primarily to rule out prescriptions in very low risk cases. The present study employed only one very low risk case and identified a prescription rate of 11%; further work is needed to assess the stability of this estimate in a larger and more varied set of very low risk cases.

While GPs overprescribed relative to STARWAVe guidelines, the rate of prescription identified here (15% across cases) is lower than that observed in other studies. For example, Hay *et al* identified a rate of 37% in their prospective cohort study of children presenting to the GP with cough.[12] Notably, present work included few high risk presentations (13%), but high risk presentations were likewise infrequent in the study by Hay *et al* (3%).[12] If our finding is reflective of real-world practice, then this reduced rate of prescribing is promising. However, it could also reflect the limitations of our vignettes, which ignored the complex interpersonal (doctor-patient) dynamics that are known to influence prescribing behaviour.[13–15 27 36] For example, prescription likelihood is increased by perceived pressure from patients/parents to prescribe;[14 27 32 37 38] by the desire to maintain good relationships with patients/parents;[13 39 40] by fear of medicolegal problems;[13 15 40] and by time pressure.[13 14 38–40] Importantly, these factors can be incorporated into clinical vignettes, as demonstrated by Sirota and colleagues; these authors found that prescriptions were twice as likely when patient pressure for antibiotics was present (vs absent) from a clinical vignette.[27] On the one hand, it is a limitation of our vignettes that these interpersonal factors were absent; on the other, our work demonstrates that antibiotics are overprescribed *even when* these interpersonal factors are absent. It is worrying that so many GPs considered antibiotics to be the most appropriate course of action, not simply the most expedient one. Qualitative research may be useful to understand why GPs prescribed to patients that they deemed to be low or medium risk, in the absence of any interpersonal pressure to do so.

Data for this study were collected in conjunction with another project, which limited the number of STARWAVe variables that we could manipulate. A comprehensive investigation of all seven STARWAVe variables would undoubtedly return new and valuable insights. Future investigations might also treat the continuous STARWAVe variables (age and illness duration) as continuous (not binary), to test the generalisability of the trends identified here.

A second consequence of collecting data in conjunction with another project is that the antibiotics vignettes (n=8) were interspersed among many cancer-related vignettes (n=48). We cannot exclude the possibility that the cancer vignettes influenced performance on the antibiotics task. For example, the cancer vignettes may have primed a hypercautious attitude (cancer being a serious, 'can't-miss' diagnosis) that lowered the threshold for intervention (prescription) in the antibiotics task. Threshold for intervention could also be lowered by response fatigue, which participants may well have experienced in assessing so many vignettes. Cognizant of this, we were careful to present the antibiotics vignettes in a random order. Random-ordering would not preclude the cancer vignettes from influencing antibiotics responding; it simply ensured that any such influence was 'spread equally' among the antibiotics vignettes.

Despite these limitations, present work sheds light on the determinants of antibiotic prescribing in child RTI presentations, bringing much-needed experimental evidence to a literature that has to date relied predominantly on self-report[13–15 33 38 40 41] and observational[28–32] data. It also speaks to the difficulties that may be encountered if STARWAVe is provided as a decision aid to GPs. First, GPs' classification of risk in this study was largely incompatible with STARWAVe's; GPs consistently chose lower risk than STARWAVe would suggest. Still, they prescribed more frequently than STARWAVe risk classification would support. Presenting GPs with STARWAVe's risk classification will likely exacerbate prescribing (since GPs overprescribed with their own, lower classifications of risk). Presenting them with a recommendation may also be ineffective, unless the recommendation is accompanied by an explanation. Explaining the recommendation in terms of the variables that increase/decrease a child's risk of hospitalisation may be a way forward and enable GPs to understand why their own intuitive decision might differ from the recommendation. Identifying the factors that are likely to be misinterpreted by GPs is important when explaining the rationale behind recommendations.

**Acknowledgements** The authors gratefully acknowledge the support of the National Institute for Health Research Clinical Research Network (NIHR CRN) in recruiting GPs for the study and infrastructure support from the NIHR Imperial Patient Safety Translational Research Centre and the NIHR Imperial Biomedical Research Centre.

**Contributors** All authors contributed to the design of the study. MN performed the data collection; MN and OK performed the data analysis. MN drafted the manuscript; OK and BD provided critical revision and approved the final version.

**Funding** This research was supported by the National Institute for Health Research (NIHR) Imperial Patient Safety Translational Research Centre. The views expressed are those of the authors and not necessarily those of the NHS, the NIHR or the Department of Health and Social Care. The funders had no role in the study design; in the collection, analysis and interpretation of the data; in the writing of the report; and in the decision to submit the paper for publication.

**Competing interests** Dr MN, Dr BD and Dr OK report grants from the NIHR Imperial Patient Safety Translational Research Centre, during the conduct of the study.

**Patient and public involvement** Patients and/or the public were not involved in the design, or conduct, or reporting, or dissemination plans of this research.

**Patient consent for publication** Not required.

**Ethics approval** Ethics approval for this study was obtained from the Health Research Authority (reference number 18/HRA/0021) and research sponsorship was provided by Imperial College London (JRO reference 17IC3882). All aspects of the study were conducted in the UK in 2018.

**Provenance and peer review** Not commissioned; externally peer reviewed.

**Data availability statement** The data are available in a public, open access repository (the Open Science Framework) under a CC-By Attribution 4.0 International Licence: https://osf.io/r3ype/

**ORCID iD**

Martine Nurek http://orcid.org/0000-0002-4252-4692

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
