## [Reviewer comments · BMJ Open]

ARTICLE DETAILS

TITLE (PROVISIONAL)	Risk assessment and antibiotic prescribing decisions in children presenting to UK primary care with cough: a vignette study
AUTHORS	Nurek, Martine; Delaney, Brendan; Kostopoulou, Olga

VERSION 1 – REVIEW

REVIEWER	Steinar Hunskaar University of Bergen, Norway
REVIEW RETURNED	30-Nov-2019

GENERAL COMMENTS	This is a vignette based study on GPs risk assessment for hospitalization and willingness to prescribe antibiotics for children with cough. The risk assessment is based on STARWAVE, a validated risk assessment tool with seven variables, that can be used as a clinical prediction rule (CPR). Eight cases were presented for 252 GPs, altogether 1008 cases were analysed. The study shows underestimated risk for hospitalization and overprescription of antibiotics, when STARWAVE was the gold standard. In many ways this is an impressive study, well conducted, good response rate, systematic variation of the variables, and an adequate analysis plan. The paper is well written and easy to follow. There are, however, some important points that the authors should consider in order to improve the paper. The authors seem very determined in using the STARWAVE as the gold standard. There is a reasonable argument for using this as a CRP. But in my view the authors argue that the GPs are always “the bad guys” in a way that hide for a good discussion about the reasons for the differences. Such explanations can of course be of several types. There can be methodological challenges with the study design, there could be problems with validity and relevance of the vignettes, there could be alternative ways of analysis, and of course there could be lack of correct medical knowledge among the GPs. I feel that in the present paper that the authors rush to the latter conclusion, and that the discussion of the results and the limitations of the study are not adequately addressed. The eight vignettes were presented within a study with quite another purpose, namely a study on suspected cancer. This was the main purpose and focus, also indicated by the fact that the RTI cases were 2 out of 26 in the first round and 2 out of 26 in the second round. The possible effect and impact of this is not
--

	discussed. The vignettes are very similar, and the “keys” to difference may not be easy to grasp in a hurry when answering a study like this. The clinical relevance may therefore be questionable. However, I acknowledge the vignette methodology as an approach to this kind of research, but limitations must be adequately discussed. The association between risk and antibiotic prescription is also not adequately discussed. The original STARWAVE publication states that “Clinical characteristics can distinguish children at very low, normal, and high risk of future hospital admission for respiratory tract infection and could be used to reduce antibiotic prescriptions in primary care for children at very low risk.” The original conclusion is thus the use as a “rule out” prescription when the risk is very low. There is only one vignette with very low risk in the study (no 1), and only 1% of the GPs assess this as high risk and 11% would prescribe antibiotics. The 11% may seem unexpectedly high based on the risk assessment, but is anyway a very low percentage. The overall percentage of prescription was 15%, much lower than in the STARWAVE cohort (>30%). In 5 of 8 vignettes the prescription rate is less than 15%, and would represent a major positive step if it was in the real world. The factor triggering prescribing seems to be length of illness (6 days) (shown in Table 3), opposite of the data from the CPR. The original CPR paper measured hospitalizations after 30 days, and many of them occurred after 3-6 days. There should be a discussion about the possible reasonable approach among the GPs when they await prescription for some days, and that this in fact can be part of the strategy if the alternative is to prescribe early in the illness. Also, the size of original high risk group was only 0.9%, which is the epidemiological landscape the GPs navigate in. In the present vignette study the high risk group was 12.5% (126/1008). The major misclassification was that GPs tended to conclude with very low risk when the “correct” answer was medium risk. The possible (small) impact of this in daily clinical practice is not adequately discussed, and again the main focus is on the “mistaken” GPs in this matter. In conclusion, the authors have performed a relevant and possibly important study in order to shed some light on GPs’ risk assessment and prescription in children with RTIs. The methodology and the results of the paper is however, not adequately discussed, and the presented conclusions and implications are too narrow in perspective.
--	--

REVIEWER	Thea Brennan-Krohn Boston Children's Hospital, United States
REVIEW RETURNED	12-Feb-2020

GENERAL COMMENTS	Comments: This is a clearly-written, well-presented study that takes an interesting approach to evaluation of clinical decisions around antibiotic prescribing. My two main comments concern potential qualifications that should be considered in evaluating the results, and which I think deserve comment in the manuscript.
---

	1) The fact that the vignettes evaluated in this study were interspersed among a much larger set of vignettes relating to referral for suspected cancer seems like it might have influenced responses. For example, if throughout most of the questionnaire respondents were primed to be considering a very serious, can't-miss diagnosis (cancer), it seems that they might have had a lower threshold for intervention (i.e. prescribing antibiotics). On the other hand, they might also have underestimated the likelihood of hospitalization in children presenting with cough if they were inadvertently comparing these vignettes to vignettes involving children in more urgent need of referral/hospitalization. I don't think there's any way to sort out these factors with the data as obtained from the current study, and I don't think it invalidates the results, but I do think it deserves some discussion. I would also like to know whether respondents were specifically alerted when they were going to be reading a vignette that belonged to the current study rather than to the cancer study. 2) In the second paragraph on page 17, the authors discuss respondents "underprescribing" for the patient in the high risk vignette, but the original STARWAVE validation article (PMID 28490554) does not prove or argue that all children in the high risk group need an antibiotic prescription – indeed, they note that only 26.9% of children who were actually hospitalized had a discharge diagnosis suggestive of a bacterial infection. The focus of the original validation seems to have been entirely on avoiding antibiotics in children who are not at high risk.
--	--

VERSION 1 – AUTHOR RESPONSE

Reviewer 1 (Steinar Hunskaar)

This is a vignette based study on GPs risk assessment for hospitalization and willingness to prescribe antibiotics for children with cough. The risk assessment is based on STARWAVE, a validated risk assessment tool with seven variables, that can be used as a clinical prediction rule (CPR). Eight cases were presented for 252 GPs, altogether 1008 cases were analysed.

The study shows underestimated risk for hospitalization and overprescription of antibiotics, when STARWAVE was the gold standard.

In many ways this is an impressive study, well conducted, good response rate, systematic variation of the variables, and an adequate analysis plan. The paper is well written and easy to follow. There are, however, some important points that the authors should consider in order to improve the paper.

1) The authors seem very determined in using the STARWAVE as the gold standard. There is a reasonable argument for using this as a CRP. But in my view the authors argue that the GPs are always "the bad guys" in a way that hide for a good discussion about the reasons for the differences. Such explanations can of course be of several types. There can be methodological challenges with the study design, there could be problems with the present use of the CPR, there could be problems with validity and relevance of the vignettes, there could be alternative ways of analysis, and of course there could be lack of correct medical knowledge among the GPs. I feel that in the present paper that the authors rush to the latter conclusion, and that the discussion of the results and the limitations of the study are not adequately addressed.

Response: thank you for pointing this out; we do not wish to unfairly criticise GPs. Indeed, there are many reasons why GPs may have deviated from the STARWAVE rule, and we now address these in

the Discussion (p18: "This is the first...").

2) The eight vignettes were presented within a study with quite another purpose, namely a study on suspected cancer. This was the main purpose and focus, also indicated by the fact that the RTI cases were 2 out of 26 in the first round and 2 out of 26 in the second round. The possible effect and impact of this is not discussed. The vignettes are very similar, and the "keys" to difference may not be easy to grasp in a hurry when answering a study like this. The clinical relevance may therefore be questionable. However, I acknowledge the vignette methodology as an approach to this kind of research, but limitations must be adequately discussed.

Response: we agree with the reviewer's comment. We have added a paragraph to the Discussion that addresses the potential effects of the cancer-related vignettes on the antibiotics task (p21: "A second consequence of...").

3) The association between risk and antibiotic prescription is also not adequately discussed. The original STARWAVE publication states that "Clinical characteristics can distinguish children at very low, normal, and high risk of future hospital admission for respiratory tract infection and could be used to reduce antibiotic prescriptions in primary care for children at very low risk." The original conclusion is thus the use as a "rule out" prescription when the risk is very low. There is only one vignette with very low risk in the study (no 1), and only 1% of the GPs assess this as high risk and 11% would prescribe antibiotics. The 11% may seem unexpectedly high based on the risk assessment, but is anyway a very low percentage.

Response: the small number of very low risk vignettes in this study (n = 1) is indeed a limitation, not only because "very low risk" is the primary group of interest in the STARWAVE model, but also because it is the most common risk presentation encountered by GPs. We now address this in the Discussion (p19: "A more representative set ..."). We also highlight the need for a more comprehensive investigation of prescribing patterns in very low risk cases.

4) The overall percentage of prescription was 15%, much lower than in the STARWAVE cohort (>30%). In 5 of 8 vignettes the prescription rate is less than 15%, and would represent a major positive step if it was in the real world. The factor triggering prescribing seems to be length of illness (6 days) (shown in Table 3), opposite of the data from the CPR. The original CPR paper measured hospitalizations after 30 days, and many of them occurred after 3-6 days. There should be a discussion about the possible reasonable approach among the GPs when they await prescription for some days, and that this in fact can be part of the strategy if the alternative is to prescribe early in the illness.

Response: we have added this to the Discussion (p18: "This is the first...").

5) Also, the size of original high risk group was only 0.9%, which is the epidemiological landscape the GPs navigate in. In the present vignette study the high risk group was 12.5% (126/1008).

Response: we have added this to the Discussion (p18: "This is the first...").

6) The major misclassification was that GPs tended to conclude with very low risk when the "correct" answer was medium risk. The possible (small) impact of this in daily clinical practice is not adequately discussed, and again the main focus is on the "mistaken" GPs in this matter.

Response: we do not entirely agree that this was "the major misclassification". It was certainly the most common misclassification, but medium risk cases were the most common type of case. In fact, underestimation of risk was twice as common in high risk presentations (90%, 114/126) than in

medium risk ones (46%, 345/756), but there was perhaps less opportunity to observe this misclassification because high risk presentations were relatively infrequent (n = 126 high risk presentations vs. n = 756 medium risk presentations; see Table 1 on p13). We realise, however, that the predominance of medium risk presentations limits the clinical significance of the findings: in reality, only 30% of child RTI presentations are medium risk. (This predominance was an unavoidable consequence of our fractional factorial design, which required that specific STARWAVE characteristics be present in specific vignettes.) We now acknowledge this in the Discussion, and suggest that future research employ a more representative selection of vignettes (p19: “A more representative set...”).

In conclusion, the authors have performed a relevant and possibly important study in order to shed some light on GPs’ risk assessment and prescription in children with RTIs. The methodology and the results of the paper is however, not adequately discussed, and the presented conclusions and implications are too narrow in perspective.

Reviewer 2 (Thea Brennan-Krohn)

Comments: This is a clearly-written, well-presented study that takes an interesting approach to evaluation of clinical decisions around antibiotic prescribing. My two main comments concern potential qualifications that should be considered in evaluating the results, and which I think deserve comment in the manuscript.

1) The fact that the vignettes evaluated in this study were interspersed among a much larger set of vignettes relating to referral for suspected cancer seems like it might have influenced responses. For example, if throughout most of the questionnaire respondents were primed to be considering a very serious, can’t-miss diagnosis (cancer), it seems that they might have had a lower threshold for intervention (i.e. prescribing antibiotics). On the other hand, they might also have underestimated the likelihood of hospitalization in children presenting with cough if they were inadvertently comparing these vignettes to vignettes involving children in more urgent need of referral/hospitalization. I don’t think there’s any way to sort out these factors with the data as obtained from the current study, and I don’t think it invalidates the results, but I do think it deserves some discussion.

Response: we agree with the reviewer’s comment. We have added a paragraph to the Discussion, addressing the potential influence of the cancer-related vignettes on antibiotics responding (p21: “A second consequence of...”). We are currently in the process of designing a study that will disentangle these issues.

2) I would also like to know whether respondents were specifically alerted when they were going to be reading a vignette that belonged to the current study rather than to the cancer study.

Response: Yes, each antibiotics vignette was preceded by the following text: “We understand that this is somewhat monotonous, so here is something quite different to help you re-engage attention.” We have added this to the Methods section of the manuscript (p9: “Interested participants were e-mailed...”).

3) In the second paragraph on page 17, the authors discuss respondents “underprescribing” for the patient in the high risk vignette, but the original STARWAVE validation article (PMID 28490554) does not prove or argue that all children in the high risk group need an antibiotic prescription – indeed, they note that only 26.9% of children who were actually hospitalized had a discharge diagnosis suggestive of a bacterial infection. The focus of the original validation seems to have been entirely on avoiding antibiotics in children who are not at high risk.

Response: we agree and have modified the paragraph accordingly (p17: “Interestingly, the one patient...”).

VERSION 2 – REVIEW

REVIEWER	Steinar Hunskaar University of Bergen, Norway
REVIEW RETURNED	29-Mar-2020

GENERAL COMMENTS	The Authors have revised their paper within a framework of understanding my comments. The revision is fully acceptable and the ne discussion points well phrased
--

REVIEWER	Thea Brennan-Krohn Beth Israel Deaconess Medical Center, Boston, MA, USA
REVIEW RETURNED	10-Mar-2020

GENERAL COMMENTS	The authors have added thoughtful discussion that addresses my concerns and (in my opinion) those of the other reviewer as well.
--